# Combination of Zearalenone and Deoxynivalenol Induces Apoptosis by Mitochondrial Pathway in Piglet Sertoli Cells: Role of Endoplasmic Reticulum Stress

**DOI:** 10.3390/toxins15070471

**Published:** 2023-07-21

**Authors:** Sirao Hai, Jiawen Chen, Li Ma, Chenlong Wang, Chuangjiang Chen, Sajid Ur Rahman, Chang Zhao, Shibin Feng, Jinjie Wu, Xichun Wang

**Affiliations:** 1College of Animal Science and Technology, Anhui Agricultural University, Hefei 230036, China; haisirao@stu.ahau.edu.cn (S.H.); chenjiawen@stu.ahau.edu.cn (J.C.); mali@stu.ahau.edu.cn (L.M.); wangchenlong@stu.ahau.edu.cn (C.W.); 22720207@stu.ahau.edu.cn (C.C.); dr_sajid226@yahoo.com (S.U.R.); chang_zhao@ahau.edu.cn (C.Z.); luyifsb@126.com (S.F.); 2Department of Food Science and Engineering, School of Agriculture and Biology, Shanghai Jiao Tong University, Shanghai 200240, China; 3Anhui Province Engineering Laboratory for Animal Food Quality and Bio-Safety, Hefei 230036, China

**Keywords:** zearalenone, deoxynivalenol, piglet Sertoli cells, endoplasmic reticulum stress, mitochondrial pathway apoptosis

## Abstract

Zearalenone (ZEA) and deoxynivalenol (DON) are widely found in various feeds, which harms livestock’s reproductive health. Both mitochondria and endoplasmic reticulum (ER) can regulate cell apoptosis. This study aimed to explore the regulatory mechanism of endoplasmic reticulum stress (ERS) on ZEA- combined with DON-induced mitochondrial pathway apoptosis in piglet Sertoli cells (SCs). The results showed that ZEA + DON damaged the ultrastructure of the cells, induced apoptosis, decreased mitochondrial membrane potential, promoted the expression of cytochrome c (CytC), and decreased the cell survival rate. Furthermore, ZEA + DON increased the relative mRNA and protein expression of *Bid*, *Caspase-3*, *Drp1*, and *P53*, while that of *Bcl-2* and *Mfn2* declined. ZEA + DON was added after pretreatment with 4-phenylbutyric acid (4-PBA). The results showed that 4-PBA could alleviate the toxicity of ZEA + DON toward SCs. Compared with the ZEA + DON group, 4-PBA improved the cell survival rate, decreased the apoptosis rate, inhibited CytC expression, and increased mitochondrial membrane potential, and the damage to the cell ultrastructure was alleviated. Moreover, after pretreatment with 4-PBA, the relative mRNA and protein expression of *Bid*, *Caspase-3*, *Drp1*, and *P53* were downregulated, while the relative mRNA and protein expression of *Bcl-2* and *Mfn2* were upregulated. It can be concluded that ERS plays an important part in the apoptosis of SCs co-infected with ZEA-DON through the mitochondrial apoptosis pathway, and intervention in this process can provide a new way to alleviate the reproductive toxicity of mycotoxins.

## 1. Introduction

ZEA and DON, mycotoxins derived from *Fusarium* fungi, are commonly detected in mold-infested grains like corn, wheat, barley, oats, and sorghum [1]. One study found that individual contamination with ZEA and DON exceeded 96.9% and 96.4%, respectively, in feeds originating from various regions in China between 2018 and 2020. Furthermore, a significant majority of feed ingredients (over 81.5%) and complete feeds (about 95.7%) were found to be co-infected with different combinations of these mycotoxins [2]. ZEA is similar in structure to estrogen, so the main toxic effects are reproductive toxicity, which is mainly manifested by reduced reproductive performance and production performance [3]. Research has indicated that it can cause oxidative damage and trigger apoptosis in porcine germ cells through the Nrf2 signaling pathway [4]. In addition, ZEA can activate the signaling pathways associated with ferroptosis in mouse testes, which can induce spermatogenesis impairment in mice [5]. DON is one of the most representative mycotoxins in the monoterpene B group. It has been reported that DON exposure may destroy the tight junctions of bovine mammary alveolar cells (MAC-T), leading to cell apoptosis [6]. It also induces IPEC-J2 cell death through oxidative stress and ferroptosis signaling pathways [7]. Under natural circumstances, ZEA and DON often coexist in animal feed [8]. When the feed of animals is contaminated with ZEA and DON, they mainly damage the reproductive system, resulting in reduced production performance and reduced fecundity, resulting in huge economic losses [9]. SCs are the predominant somatic cells in the vas deferens and play a crucial role in regulating spermatogenesis. As each SC can support a fixed number of germ cells, the number of functional SCs directly impacts the overall production capacity of sperm. ZEA and DON can significantly cause oxidative damage and induce apoptosis in SCs, and their combined toxicity is greater than their individual toxicities [10]. Feeding ZEA and DON can significantly reduce the oocyte quality of sows [11].

The interaction between mitochondria and the ER is a meticulously regulated process, culminating in the formation of specialized areas called mitochondria-associated endoplasmic reticulum membranes (MAMs) [12]. Our previous research found that ZEA increases the distance of MAM and disrupts its structure and function through the mitochondrial pathway/ERS pathway, ultimately leading to increased intracellular Ca^2+^ levels. Pretreatment with a mitochondrial division inhibitor (Mdivi-1)/4-PBA alleviated the abnormal protein expression of each pathway induced by ZEA in MAM [13,14]. However, the specific effects of MAM regulation on apoptosis indicators related to the mitochondrial pathway were not investigated. When stimulated by apoptosis-inducing factors, phosphoric acidic cluster sorting protein 2 (PACS-2) binds to Bid, a dephosphorylated BH3 domain apoptotic protein, and drives the transport of apoptotic signals from the ER to mitochondria [15]. Meanwhile, Fis interacts with Bap31, facilitating its conversion into pro-apoptotic Bap31 via procaspase-8, thereby transmitting apoptotic signals from mitochondria to the ER [16]. However, there are few studies on whether ERS regulates mitochondria-mediated apoptosis in SCs exposed to ZEA and DON.

In this study, SCs were used as a model, 4-PBA was added to inhibit ERS, and the regulatory mechanism of ERS on mitochondrial pathway apoptosis in SCs exposed to ZEA and DON was studied to lay the theoretical groundwork for developing strategies to address the reproductive toxicity of ZEA and DON.

## 2. Results

### 2.1. Effects of ZEA + DON and 4-PBA on Cell Viability

As depicted in Figure 1A, cell survival decreased dose-dependently with increasing ZEA and DON concentrations compared with the CON group. When the concentrations of ZEA and DON were 30 μM and 1.2 μM, respectively, the combination caused a cell survival rate of nearly 50%.

As shown in Figure 1B, when the concentration of 4-PBA was 1.5 μM, it had no harmful effects on cells and significantly alleviated the harmful effects of ZEA and DON on cells.

### 2.2. Effects of ZEA + DON and 4-PBA on Cell Growth

Living cells were stained green by AM, and dead cells or cells with damaged cell membranes were labeled red by PI. Figure 2A shows that green fluorescence in the CON and P groups was uniformly distributed and strong. When cells were infected with ZEA + DON, red fluorescence was enhanced. Following pretreatment with 4-PBA, red fluorescence was observed to decrease significantly, while green fluorescence showed an increase.

### 2.3. Effects of ZEA + DON and 4-PBA on Cell Ultrastructure

As illustrated in Figure 2B, the cells in the CON group appeared plump, the mitochondrial cristae were visible, the ER was more abundant, and the structure was intact. In the P group, the cells maintained a plump appearance, and neither mitochondrial nor ER structures showed any signs of damage. In the Z + D group, the cells appeared broken, the chromatin was positioned near the nuclear membrane, the nuclear membrane was damaged, the ER cristae were ruptured, the number of mitochondria was reduced, and swelling and vacuolar degeneration occurred. Compared with the Z + D group, the mitochondrial structure of group Z + D + P cells was clearer and the ER rupture was improved.

### 2.4. Effects of ZEA + DON and 4-PBA on Apoptosis Rate

In Figure 3A,B, it can be observed that the apoptosis rate in the CON group was 0.092%. However, in the Z + D group, the apoptosis rate was significantly higher at 24.5% (*p* < 0.01). The apoptosis rate in the P group was not significantly different from that in the CON group (*p* > 0.05). The Z + D + P group exhibited a significant decrease in the apoptosis rate compared to the Z + D group (*p* < 0.01).

### 2.5. Effects of ZEA + DON and 4-PBA on Mitochondrial Membrane Potential

Figure 3C,D illustrate that the mitochondrial membrane potential (MMP) of the Z + D group was significantly decreased compared to the CON group (*p* < 0.01). However, there was no significant difference in the MMP of the P group compared to the CON group (*p* > 0.05). On the other hand, the MMP of the Z + D + P group was significantly increased compared to the Z + D group (*p* < 0.01).

### 2.6. Effects of ZEA + DON and 4-PBA on Gene Expression

In Figure 4, the relative mRNA expression levels of *Bid*, *Caspase-9*, *Drp1*, and *P53* were significantly increased (*p* < 0.01) in the Z + D group compared to the CON group, while the relative mRNA expression levels of *Bcl-2* and *Mfn2* were significantly decreased (*p* < 0.01). Compared with the CON group, there were no significant changes in the relative mRNA expression of each gene in the P group (*p* > 0.05). On the other hand, the relative mRNA expression levels of *Bid*, *Caspase-9*, *Drp1*, and *P53* were significantly decreased in the Z + D + P group compared to the Z + D group (*p* < 0.01 or *p* < 0.05), and the relative mRNA expression levels of *Bcl-2* and *Mfn2* were significantly increased in the Z + D + P group (*p* < 0.01 or *p* < 0.05).

### 2.7. Effects of ZEA + DON and 4-PBA on Protein Expression

Figure 5 reveals that the protein expression levels of *Bid*, *Caspase-3*, *Drp1*, and *P53* were significantly increased (*p* < 0.01) in the Z + D group compared to the CON group, whereas the protein expression levels of *Bcl-2* and *Mfn2* were significantly decreased (*p* < 0.01). In contrast, the protein expression levels of *Bid*, *Caspase-9*, *Drp1*, and *P53* were significantly decreased (*p* < 0.01) in the Z + D + P group compared to the Z + D group, while the protein expression levels of *Bcl-2* and *Mfn2* were significantly increased (*p* < 0.01 or *p* < 0.05).

### 2.8. Effects of ZEA + DON and 4-PBA on CytC Expression and Distribution

Figure 6 demonstrates that the CytC fluorescence was very weak in the CON group and P group, whereas it was strongly positive and predominantly localized around the nucleus in the Z + D group. Furthermore, the immunostaining of CytC in the Z + D + P group was relatively weak compared to the Z + D group.

## 3. Discussion

### 3.1. ZEA and DON Induce Cytotoxicity and Apoptosis through Mitochondrial-ER Pathway

At present, mycotoxins are one of the main threats to animal feed safety. Due to the lack of studies on the additive toxicity and harm of mycotoxins, the mycotoxin hazard in agricultural production is often ignored. ZEA and DON were the most sensitive to swine. Research has demonstrated that ZEA and DON induce cell apoptosis via the mitochondrial pathway and induce Cyt C release in GC-1 spermatogonia (spg) cells, thereby causing mitochondrial pathway apoptosis [17]. ZEA stimulates mitochondrial fission and leads to apoptosis in endometrial stromal cells (ESCs) by activating the JNK/Drp1 pathway [18]. Prior research conducted in our laboratory has demonstrated that ZEA has the ability to harm cellular structures, hinder cell growth, and initiate cellular demise alongside apoptosis in SCs [4]. The supplementation of feed with DON can have varied effects on protein synthesis and humoral and cellular immune responses in pigs, contingent on the timing, exposure, and functional immunoassay, concurrently altering the cellular structure of the liver and spleen. Moreover, reproductive changes have also been observed in pigs, and DON reduces oocyte and embryo development both in vivo and in vitro [19]. DON upregulates mitochondrial pathway apoptosis-related protein expression levels in mouse ESCs to induce apoptosis [20]. DON was found to induce epithelial vacuolation of spermatic cells, disrupt the adherens junctions of SCs established by n-cadherin, and induce oxidative stress in the testes of male mice [21]. In addition, the apoptosis rate also increased with the dose of DON. In our study, SCs were used as a model. A significant increase in the apoptosis rate of SCs was observed upon treatment with ZEA and DON. 

### 3.2. How ZEA and DON Altered Gene and Protein Expression Related to Mitochondrial Apoptotic Pathway

In granulosa cells, ZEA has been found to upregulate the expression of Bax and downregulate the expression of *Bcl-2*. These alterations promote the release of Cyt C to the cytosol, subsequently resulting in mitochondria-mediated apoptosis [22]. The indirect inhibition of mitochondrial biogenesis, mitochondrial electron transport chain function, ATP synthesis, and mitochondrial transcription and translation, followed by mitochondria-mediated apoptosis, can be induced by DON [23]. The relative expression of *Bid*, *Bcl*-2, *Caspase*-9, and *P53* and the levels of *Bid*, *Bcl*-2, *Caspase*-9, *P53*, and Cyt C proteins related to the mitochondrial apoptosis pathway were significantly upregulated, and the mitochondrial membrane potential was significantly decreased. These results suggest that ZEA combined with DON triggered SC apoptosis via the mitochondrial pathway, and the experimental results are consistent with those reported above.

### 3.3. Cell Homeostasis and Apoptosis Regulation Are Impacted by Mitochondrial Dynamics

Numerous studies have established a strong relationship between mitochondrial dynamics and apoptosis [24]. In the process of apoptosis, abundant DRP1 proteins are transferred to the outer mitochondrial membrane, promoting mitochondrial fission and ultimately resulting in the disruption of the mitochondrial network, which in turn promotes apoptosis by promoting Bax minimization and increasing outer membrane permeability (MOMP). When HEK293 was treated with 4-PBA, the level of mitochondrial fission protein (DRP1, FIS1) decreased, which was related to its protective effect on apoptosis [25]. Inhibiting ERS can impact the processes of mitochondrial fusion and fission, which consequently results in an increase in mitochondrial anti-apoptotic activity [26]. Essentially, by inhibiting ERS, the equilibrium between mitochondrial fusion and fission can be disrupted, resulting in an enhancement of the cell’s ability to resist apoptosis or programmed cell death. This is because the appropriate equilibrium between mitochondrial fusion and fission is critical for the normal functioning of mitochondria, which have a significant impact on maintaining cellular balance and regulating programmed cell death. In our experiments, after the pretreatment of SCs with 4-PBA, the results of both Western blotting (WB) and qRT-PCR indicated a remarkable decrease in the expression level of Drp1 and an increase in the expression level of *Mfn2*, suggesting that the ER regulates the mitochondrial apoptosis pathway by affecting mitochondrial dynamics. 

### 3.4. The ER’s Role in Regulating ZEA- and DON-Induced Apoptosis in SCs through the Mitochondrial Pathway in MAMs

Research has demonstrated that mitochondria and the ER can communicate apoptosis signals to each other through MAMs [27]. The mitochondrial apoptotic pathway is generally thought to be associated with excessive Ca^2+^ uptake by mitochondria. Ca^2+^ has long been acknowledged as a player in the apoptotic pathway, and the *Bcl-2* family proteins enable mitochondria to regulate Ca^2+^ concentrations, ultimately inducing apoptosis when Ca^2+^ accumulates in mitochondria. MAMs are pivotal in transferring Ca^2+^ from the ER to mitochondria. With the persistence of ERS, Ca^2+^ is released from the ER into mitochondria via MAMs, causing mitochondrial Ca^2+^ overload and ultimately activating mitochondrial pathway apoptosis [28,29]. Simultaneously, the decrease in MAM quantity impedes ER-to-mitochondria Ca^2+^ transfer, which attenuates apoptotic signaling [30]. Mitochondrial function is a key indicator of cellular health. Several critical steps constitute the mitochondrial apoptotic pathway, involving changes in mitochondrial membrane permeability, proteins exhibiting pro-apoptotic activity, and the disruption of MMP [31]. Hyperglycemia leads to a notable decline in MMP in atrial cardiomyocytes, which can be improved by inhibiting ERS [32]. Clock gene knockout can stabilize MMP and counteract apoptosis induced through the mitochondrial pathway [33]. Moreover, the release of Cyt C is considered to be a crucial characteristic of apoptosis, whereby Cyt C translocates from the mitochondrial intermembrane space to the cytoplasm via the mitochondrial permeability transition pore, initiating selective Caspase activation and ultimately resulting in the induction of apoptosis [34]. Research has revealed that DON triggers apoptosis in neural cells (PHNCs) through the apoptotic pathway mediated by mitochondria, leading to the inhibition of PHNC proliferation, as well as significant morphological, biochemical, and transcriptional modifications in line with apoptosis. Such effects include reductions in mitochondrial membrane potential, along with the liberation of Cyt C and apoptosis-inducing factor (AIF) [35]. *Mfn2* is primarily found in the outer mitochondrial membrane (OMM) and plays a vital role in the establishment and maintenance of MAM structures [13]. Localizing IFN2 through the ER has been found to induce mitochondrial fission via direct binding to Drp1 and the promotion of Drp1 oligomerization [36]. In this study, we observed a very significant decrease in MMP and significantly increased CytC expression in the Z + D group, which was significantly improved by inhibiting ERS. Our results indicate that ERS contributes to the ZEA- and DON-induced mitochondrial apoptotic pathway in SCs by influencing MMP and Cyt C expression.

Additionally, the activation and transfer of apoptosis-related factors are also involved in the interaction between mitochondria and ER. While *Bcl-2* localizes to the ER in resting cells, it is translocated to MAMs during the initial stages of apoptosis, transmitting apoptotic signals to mitochondria. *Bcl-2* is primarily located in the ER of resting cells. However, during the early stages of apoptosis, *Bcl-2* is relocated to the MAMs, which are specialized regions of the ER that interact closely with the outer mitochondrial membrane. This translocation of *Bcl-2* to the MAMs is a crucial step in the transmission of apoptotic signals from the ER to the mitochondria [27]. Once localized to the MAMs, *Bcl-2* can interact with other proteins and molecules to regulate the movement of calcium ions from the ER to the mitochondria, which can trigger MOMP and initiate apoptosis. In summary, the translocation of *Bcl-2* from the ER to the MAMs during early apoptosis is a crucial stage in the initiation of apoptotic signaling pathways that lead to cellular death. Elevated levels of CHOP and cleaved Caspase-3 are observed after mitochondrial fission, which can induce ERS to induce apoptosis [37]. The addition of Mdivi-1 inhibited the expression of the ERS markers GRP78 and C/EBP homologous protein, inhibited Caspase-3 activation, and the rate of apoptosis in hippocampal neurons [38]. In SiHa cells, Trichomonas vaginalis triggers ERS-induced mitochondrial apoptosis through the IRE1/ASK1/JNK/*Bcl-2* family protein pathways [39]. Our previous research indicates that adding 4-PBA can alleviate toxic damage and apoptosis in porcine germ cells and alleviate ERS caused by ZEA [14,40]. *Bcl-2* family proteins play a critical role in regulating the release of apoptotic factors related to mitochondria, which are mainly divided into pro-apoptotic factors and anti-apoptotic factors. *Bcl-2* acts as an anti-apoptotic factor, while Bid functions as a pro-apoptotic factor. In addition, the tumor suppressor inhibits the anti-apoptotic factor *Bcl-2* in response to DNA damage [41]. There is evidence from previous studies conducted within our research group suggesting that ZEA has the potential to cause damage to piglet SCs through the mitochondrial fission pathway, while the ER can regulate mitochondria through MAMs [13,14]. This study is based on the previous foundation of in-depth research; according to our WB and qRT-PCR results, in the Z + D group, there was a significant increase in the gene and protein expression levels of *Bid*, *P53*, and *Caspase*-9. However, inhibiting ER stress with 4-PBA led to a marked decrease in the gene and protein expression levels of *Bid* and *P53*. In the Z + D group, the gene and protein expression levels of the anti-apoptotic factor *Bcl*-2 were significantly reduced. However, the addition of 4-PBA resulted in a notable increase in the gene and protein expression levels of *Bcl*-2. These results suggest that ERS can affect apoptotic factors related to the mitochondrial apoptosis pathway and induce apoptosis in SCs that is triggered by the combination of ZEA and DON.

## 4. Conclusions

In conclusion, the inhibition of ERS alleviates ZEA- and DON-induced mitochondrial pathway apoptosis. The results suggest that the ER, acting as the upstream mediator of the mitochondrial apoptotic pathway, participates in the apoptosis of SCs triggered by ZEA and DON. Intervention in this procedure can serve as a novel concept for the relief of mycotoxin reproductive toxicity.

## 5. Materials and Methods

### 5.1. Chemical and Reagents

The piglet SCs was obtained from the cell bank of BLUEFBIO (Shanghai, China). ZEA and DON were obtained from Sigma-Aldrich (St. Louis, MO, USA). 4-PBA was obtained from Target Molecule (Boston, MA, USA). Dulbecco’s modified eagle medium (DMEM) high-glucose medium was purchased from Basalmedia (Shanghai, China). Fetal Bovine Serum (FBS) was purchased from Excell Bio (Suzhou, China). Cell Counting Kit-8 (CCK-8) and bicinchoninic acid kit were purchased from Sparkjade (Shandong, China). Supper RIPA Lysis Buffer was purchased from Coolaber (Beijing, China). The apoptosis detection kit containing Annexin V-fluorescein isothiocyanate (FITC) and propidium iodide (PI) was obtained from Yeasen Biotech Co, Ltd. (Shanghai, China). The mitochondrial membrane potential assay kit with JC-1 and the calcein/PI cell viability/cytotoxicity assay kit were obtained from Beyotime Biotechnology. The digital slide scanner was from 3DHISTECH (Budapest, Hungary). Antibodies to ACTIN, *Bcl-2*, *Bid Caspase-9*, *Drp1*, *MFN2*, and *P53* were purchased from Servicebio (Wuhan, China). The anti-Cyt C antibodies were purchased from HuaBio (Hangzhou, China).

### 5.2. Cell Culture and Treatments

Cells were treated with varying concentrations of ZEA (0, 15, 30, 45, 60, 80 μM) and DON (0, 0.4, 0.8, 1.2, 1.8, 2.4, 4.8 μM) for 24 h. Prior to exposure to ZEA and DON, cells were pretreated with 4-PBA for 2 h. In subsequent experiments, 1.5 μM 4-PBA and a combination of 30 μM ZEA and 1.2 μM DON were used. The details of the methods of cell culture can be found in Ma et al. [13].

### 5.3. Cell Viability Assay

A total of 8 × 10^3^ cells were seeded in each well of 96-well plates and cultured for 24 h. ZEA, DON, and 4-PBA were added to the plate according to the experimental concentrations and incubated for 24 h. Subsequently, 10 μL of CCK-8 reagent was added to each well and incubated for 2 h. The cell viability was determined by measuring the absorbance at 450 nm (Thermo Mk3, Waltham, MA, USA).

### 5.4. Detection of cell Survival Status

After culturing the cells in 24-well plates for 24 h, the cells were treated with Calcein AM/PI working solution for 30 min at 37 °C. The cell viability was observed using a fluorescence optical microscope. Further details can be found in a previous study conducted by our laboratory [40].

### 5.5. Changes in Cell Ultrastructure

The cells (1.6 × 10^4^/well) were cultured in a six-well plate for 24 h, then collected and fixed with 2.5% glutaraldehyde. They were then dehydrated, cured, and sectioned before being examined for changes in cellular ultrastructure and mitochondrial morphology using transmission electron microscopy (TEM, JEM-1400, JEIL Ltd., Tokyo, Japan). For further details, please refer to the study conducted by Wang et al. [42].

### 5.6. Detection of Mitochondrial Membrane Potential

After culturing the cells, the collected cells were incubated with JC-1 working solution for 20 min at 37 °C. The mitochondrial membrane potential was detected via FACS Calibur flow cytometry (BD, Franklin Lakes, NJ, USA). For details, see our laboratory’s previous study [10].

### 5.7. Determination of Apoptotic Cells

The cells were subjected to a 24 h culture before measuring the cellular apoptosis rates using an Annexin V-FITC/PI cell apoptosis assay kit. Flow cytometry analysis (BD, Franklin Lakes, NJ, USA) was used to determine the extent of cell apoptosis. Additional information on this procedure can be found in a previous study conducted by Cao et al. [43].

### 5.8. Quantitative Real-Time Polymerase Chain Reaction (qRT-PCR)

The qRT-PCR methods employed in this study were previously reported [4]. GAPDH (glyceraldehyde-3-phosphate dehydrogenase) was used as an internal standard, and the primer sequences for this procedure were synthesized by Sangon Biotech Co., Ltd. (Shanghai, China). Please refer to Appendix A for more information on the primer sequences.

### 5.9. Determination of Protein Expression by Immunofluorescence

The cells were cultured for 24 h before being incubated with the primary antibody CytC (1:100) at 4 °C overnight, followed by anti-rabbit secondary antibody iFluor™ 488 (1:800) at 37 °C for 1 h. The fluorescent signals were examined via a digital slide scanner (Pannoramic MIDI, 3DHISTECH, Budapest, Hungary). The details were reported by our laboratory’s previous study [5].

### 5.10. Western Blotting (WB)

WB methods utilized in this study have been previously reported [4]. β-Actin (1:2000), *Bid* (1:2000), *Caspase-3* (1:2000), *Drp1* (1:2000), *P53* (1:2000), *Bcl-2* (1:2000), and *Mfn2* (1:2000) antibodies were employed.

### 5.11. Statistical Analysis

The data (*n* = 3) are presented as the mean ± standard deviation (SD), and statistical analysis was performed using one-way ANOVA followed by Tukey’s test. Statistical significance was set at *p* < 0.05. The histograms were generated using GraphPad Prism 9.0 (Graphpad Inc., San Diego, CA, USA).

## Figures and Tables

**Figure 1 toxins-15-00471-f001:**
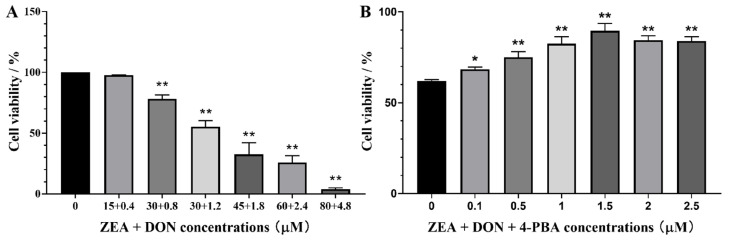
Viability of SCs upon 24 h exposure to ZEA + DON. (**A**) Effects of ZEA + DON; (**B**) effects of ZEA + DON (30 μM + 1.2 μM) upon pretreatment with 4-PBA. * and ** represent the significant difference (*p* < 0.05) and an extremely significant difference (*p* < 0.01) compared with the CON group.

**Figure 2 toxins-15-00471-f002:**
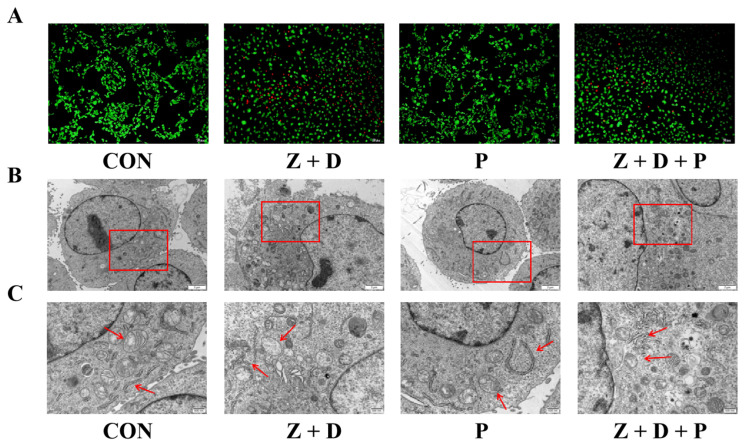
Effects of 4-PBA on the growth state induced by ZEA and DON. (**A**) The cell survival status of each group was observed with a fluorescence optical microscope (scale bar: 100 μm). Changes in cellular ultrastructure and mitochondrial morphology were observed by transmission electron microscopy; scale bars represent (**B**) 2 μm to (**C**) 500 nm. The second set of images shows the enlarged areas of the first sets (red boxes). Red arrows indicate structural changes in mitochondria and ER.

**Figure 3 toxins-15-00471-f003:**
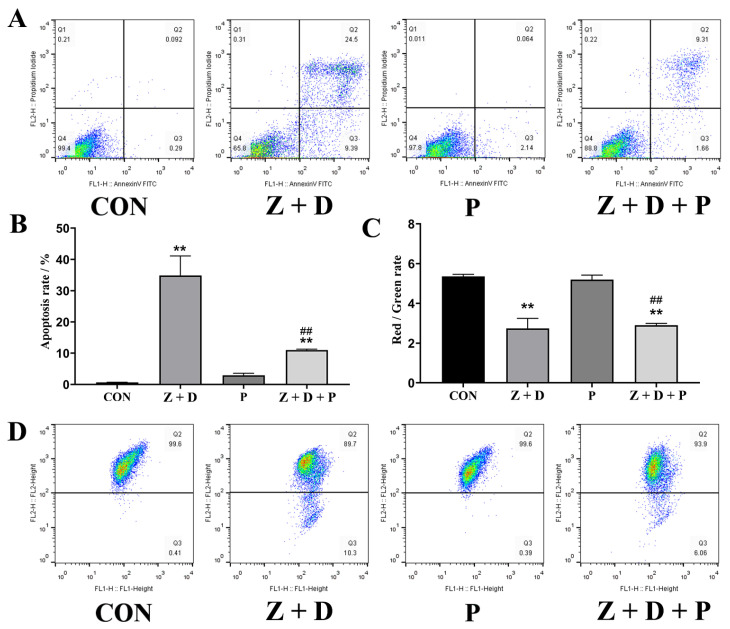
Alterations in mitochondrial membrane potential and apoptosis rate of SCs. (**A**) Annexin V-FITC/PI staining combined with flow cytometry was utilized to identify apoptotic cells. (**B**) Apoptosis rate in each group (*n* = 3). (**C**) Mitochondrial membrane potential in each group (*n* = 3). (**D**) Flow cytometry was used to detect the scatterplot of mitochondrial membrane potential. ** represent an extremely significant difference (*p* < 0.01) compared with the CON group. ## indicate extremely significant difference (*p* < 0.01) compared with Z + D group.

**Figure 4 toxins-15-00471-f004:**
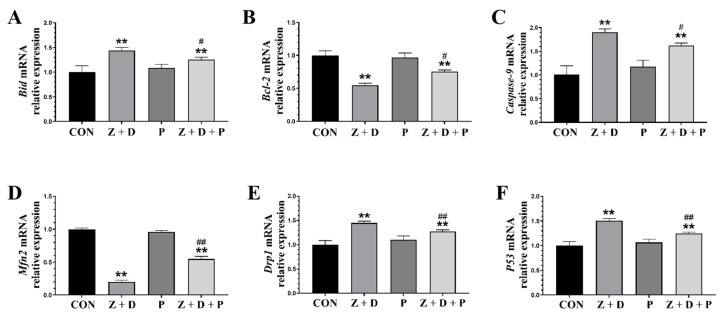
Changes in the expression of mitochondrial functional genes and mitochondrial apoptosis pathway genes. (**A**–**F**) Quantitative real-time polymerase chain reaction (qRT-PCR) analysis of *Bid*, *Caspase-9*, *Drp1*, *Bcl-2*, *Mfn2*, and *P53* mRNA expression levels (*n* = 3). ** represent an extremely significant difference (*p* < 0.01) compared with the CON group. # and ## indicate significant difference (*p* < 0.05) and extremely significant difference (*p* < 0.01) compared with Z + D group.

**Figure 5 toxins-15-00471-f005:**
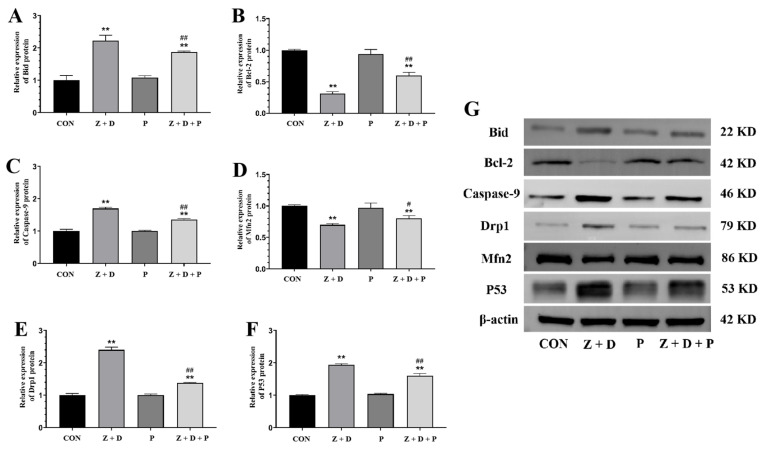
Changes in mitochondrial apoptosis pathway protein levels and mitochondrial functional proteins levels. (**A**–**G**) Western blotting analysis of *Bid*, *Caspase-9*, *Drp1*, *Bcl*-2, *Mfn2*, and *P53* protein expression levels (*n* = 3). ** represent an extremely significant difference (*p* < 0.01) compared with the CON group. # and ## indicate significant difference (*p* < 0.05) and extremely significant difference (*p* < 0.01) compared with Z + D group.

**Figure 6 toxins-15-00471-f006:**
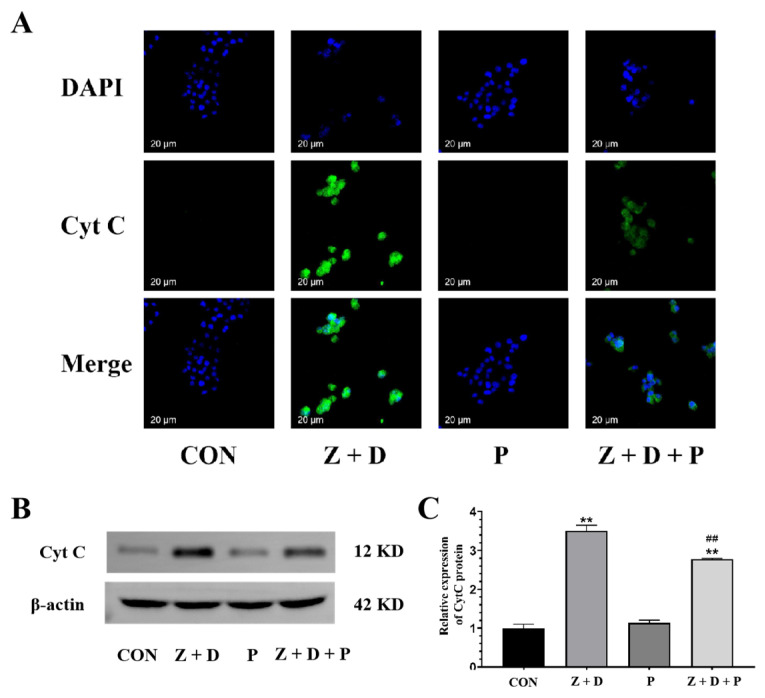
Effects of ZEA + DON on the expression and distribution of Cyt C protein by ERS. (**A**) Expression of CytC in SCs detected by immunofluorescence (scale bar: 20 µm). In the images collected, the nuclei exhibit blue fluorescence, and CytC is represented by green fluorescence. (**B**,**C**) Western blotting analysis of Cyt C protein expression levels (*n* = 3). ** represent an extremely significant difference (*p* < 0.01) compared with the CON group. ## indicate extremely significant difference (*p* < 0.01) compared with Z + D group.

## Data Availability

Not applicable.

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
