# Peer review of "Combination of Zearalenone and Deoxynivalenol Induces Apoptosis by Mitochondrial Pathway in Piglet Sertoli Cells: Role of Endoplasmic Reticulum Stress"

_toxins, 2023, doi:10.3390/toxins15070471_

Round 1
Reviewer 1 Report
Review of the manuscript entitled Endoplasmic reticulum stress involved in zearalenone and deoxynivalenol-induced mitochondrial pathway apoptosis in piglet Sertoli cells
The manuscript brings a novel data on mechanism by which combination of ZEA and DON induce apoptosis in piglet Sertoli cells trough mitochondrial pathway and how these events may be related to ER stress.
Manuscript is well organised but, in my opinion the writing of the manuscript requires improvement. Although I am not a native English speaker, I believe that English should be improved.
Specific comments are listed below.
1. The title is a little bit confusing; I suggest change into Combination of Zearalenone and Deoxynivalenol Induce Apoptosis by Mitochondrial Pathway in Piglet Sertoli Cells: Role of Endoplasmic Reticulum Stress
2. Introduction:
- Line 34: fungi should not be in italic.
- Line 35-36 (ref 2) should be deleted.
- Two papers were recently published (Ecotoxycology and Environmental Safety 2023, Toxins 2023) on ZEA-induced MAM-ER disfunction in piglet Sertoli cells related to cytotoxicity, increased Ca2+, oxidative stress and expression of specific proteins. Some of negative events were diminished by pretreatment with 4-PBA (also used in the present study) as well as with mitochondrial division inhibitor 1. It seems to me that this study is the continuation of the research on the mechanism of ZEA toxicity in MAM-ER using same model – piglet Sertoli cells. Authors should use these published studies to better explain and formulate the aim.
- Delete lines 66-73 and rewrite this part of introduction formulating the purpose of the study as continuation of previous research.
3. Results
- Why didn’t the Authors show results on single ZEA and DON cytotoxicity in comparison to ZEA+DON combination? Is that previously published elsewhere? How ZEA and DON conc. Used in combinations were chosen?
- Figure 1B: I am guessing that the same concentrations in combinations of ZEA+DON have been used as in fig 1A- this should be clearly presented in the explanation below the figure.
- Titles of all figures should be improved, by giving the general title of the figure.
- (e.g. Figure 1. Viability of Sertoli cells upon 24 h exposure to ZEA+DON. A) Effects of ZEA + DON; B) Effects of ZEA+DON upon pre-treatment with 4-PBA….
- Line 91: delete infected, use treated.
4. Discussion
- This part must be significantly improved, I suggest rewriting of the discussion. It would be easier to follow if it is divided into subheadings, e.g. 1. ZEA and DON induce cytotoxicity and apoptosis trough mitochondrial-ER pathway; 2. How ZEA and DON altered gene and protein expression related to mitochondrial apoptotic pathway.
- The Authors should discuss in connection with the previous studies that I mentioned in Introduction review.
- Lines 197-217 can be excluded.
5. Conclusion
- This part is well written.
- How this study contributed to previous results on MAM-ER in Sertoli cells?
Reviewer 2 Report
The authors has found that endoplasmic reticulum stress involved in zearalenone and deoxynivalenol-induced mitochondrial pathway apoptosis in piglet Sertoli cells. The topic is interesting. The experiment was well-designed. The following revision could improve the quality of the paper.
1. L13-14, Please specified the changes are mRNA or protein by the writting. Such as: the italic writting of abbrevation of gene for mRNA; all capital letters writting of abbrevation of protein.
2.L34,ZEA and DON. please check that all the abbreviations should with their full name at the first time appeared in the paper. Please check throughout the paper.
3. L33-42, Please add some recent references for the occurence of DON and ZEA in feeds, such as: 1) Occurrence and Exposure Assessment of Major Mycotoxins in Foodstuffs from Algeria, toxins, 2023; 2) Occurrence of Aflatoxin B1, deoxynivalenol and zearalenone in feeds in China during 2018–2020, Journal of Animal Science and Biotechnology, 2021.
4. L43-45, please add some new reference about ZEN and DON induced the novel cell deaths. Besides the apoptosis, there are ferroptosis etc. 1) Effect of Zearalenone-Induced Ferroptosis on Mice Spermatogenesis, Animals, 2022; 2) Ferroptosis is involved in deoxynivalenol-induced intestinal damage in pigs. Journal of Animal Science and Biotechnology. 2023.
5. Figures, please use different letters to express the difference between the groups; add the replicates n=? in the figure legends.
6. Figure 5, please check the Figure 5 G and H, the b-actin are the same for the two different figures. Generally, each target protein band should have thier own b-actin band.
7.L294, it is better to write the conclusion as one paragraph.
8. L324, do not write only one sentence as a paragraph.
9. L35, Quantitative real-time polymerase chain reaction (qRT-PCR), these informatin canbe submitted as online supplemental material.
10. L380, please check the references and make sure there are following the journal style.
Reviewer 3 Report
1- Abbreviations must be included when they are mentioned the 1st time, for example: 4-PBA. line 14, but there are more all over the text, specially in the Discussion section.
2- No need to include a supplementary file with the primers list, these are already described in the paper in table 1.
3- Material and Methods section with more details.
4- Discussion/Conclusion: these compounds affect reproductive cells, should include some discussion on the possible effects of 4-PBA on animal reproduction.
Round 2
Reviewer 1 Report
The authors accepted the suggestions and made corrections. I believe the mansuscript has been improved and can be accepted for publishing in Toxins. It is necessary to correct typing errors, spaces between text and brackets with references, Fusarium should be in italic.
Author Response
Thanks to your careful review. We have corrected the typing errors in the manuscript, increased the spaces between the text and the reference brackets, and italicized the Fusarium.